# Individualized machine-learning-based clinical assessment recommendation system

Devin Setiawan[1*], Yumiko Wiranto[2], Jeffrey M. Girard[2], Amber Watts[2], Arian Ashourvan[2*]

1 Department of Electrical Engineering and Computer Science, The University of Kansas, Lawrence, Kansas, United States of America, 2 Department of Psychology, The University of Kansas, Lawrence, Kansas, United States of America

* devinryandi.s@ku.edu (DS); ashourvan@ku.edu (AA)

## Abstract

Traditional clinical assessments often lack individualization, relying on standardized procedures that may not accommodate the diverse needs of patients, especially in early stages where personalized diagnosis could offer significant benefits. We aim to provide a machine-learning framework that addresses the individualized feature addition problem and enhances diagnostic accuracy for clinical assessments. Individualized Clinical Assessment Recommendation System (iCARE) employs locally weighted logistic regression and Shapley Additive Explanations (SHAP) value analysis to tailor feature selection to individual patient characteristics. Evaluations were conducted on synthetic and real-world datasets, including early-stage diabetes risk prediction and heart failure clinical records from the UCI Machine Learning Repository. We compared the performance of iCARE with a Global approach using statistical analysis on accuracy and area under the ROC curve (AUC) to select the best additional features. The iCARE framework enhances predictive accuracy and AUC metrics when additional features exhibit distinct predictive capabilities, as evidenced by synthetic datasets 1–3 and the early diabetes dataset. Specifically, in synthetic dataset 1, iCARE achieved an accuracy of 0.999 and an AUC of 1.000, outperforming the Global approach with an accuracy of 0.689 and an AUC of 0.639. In the early diabetes and heart disease dataset, iCARE shows improvements of 6–12% in accuracy and AUC across different numbers of initial features over other feature selection methods. Conversely, in synthetic datasets 4–5 and the heart failure dataset, where features lack discernible predictive distinctions, iCARE shows no significant advantage over global approaches on accuracy and AUC metrics. iCARE provides personalized feature recommendations that enhance diagnostic accuracy in scenarios where individualized approaches are critical, improving the precision and effectiveness of medical diagnoses.

**Data availability statement:** The synthetic dataset generation code, along with all other code used in this study, is publicly available in a stable repository at https://doi.org/10.5281/zenodo.15299957. This repository contains the full implementation of the iCARE framework, including synthetic data generation scripts and the dataset used in all experiments which are located in the "Recreated Experiments/ ExperimentData" directory. The publicly available datasets used in this study can also be accessed individually at https://doi.org/10.24432/C5VG8H (Early Stage Diabetes Risk Prediction), https://doi.org/10.24432/C5Z89R (Heart Failure Clinical Records), and https://www.kaggle.com/datasets/johnsmith88/heart-disease-dataset (Heart Disease Dataset).

**Funding:** This work was supported by startup funding from the Department of Psychology at the University of Kansas (to AA) and the National Institutes of Health (award number R01MH125740 to JMG). The funders had no role in study design, data collection and analysis, decision to publish, or preparation of the manuscript.

**Competing interests:** The authors have declared that no competing interests exist.

## Author summary

In healthcare, the path to a diagnosis often follows a standard set of procedures. However, this "one-size-fits-all" approach can be inefficient, as the most informative next step for one person might be different for another, especially in the early stages of a disease. To solve this problem, we developed a machine-learning framework called iCARE. Our system works by learning from the health records of past patients to create a personalized model for each new individual. Based on this custom model, it then recommends which specific medical test would be most valuable to collect next to improve diagnostic accuracy. We tested our approach on medical data for conditions like diabetes and heart disease. We found that when different tests are uniquely useful for different types of patients, our personalized system improved diagnostic accuracy by 6–12% over standard methods. Our work demonstrates how machine learning can enhance the dynamic and patient-centered nature of clinical assessments.

## 1. Background

Prediction of possible outcomes and prediction of treatment impact are two important components of today's medical care and customized healthcare [1]. Clinical assessment is the ongoing process of gathering information about a patient and constructing an increasingly comprehensive conceptualization of their health and needs (e.g., for diagnosis, prognosis, or treatment planning). Machine learning (ML) approaches are commonly used for predicting and classifying diseases that are precisely utilized as an efficient tool for doctors and specialists [2]. A critical task in clinical assessment is selecting the *next* piece of information to collect about the patient to maximize information gain. Given the unique nature of each patient's condition, it is essential to recognize that there are often no one-size-fits-all solutions. This need for personalization is especially high when symptom presentation and treatment effectiveness are heterogeneous across individuals; examples include oncology, psychiatry, and the treatment of chronic diseases such as diabetes, cardiovascular disease, and neurodegenerative disorders [3–5]. A specific example in oncology would be Hepatocellular carcinoma (HCC), a primary liver cancer with an aggressive nature and that despite research, the prognosis remains unfavorable, with considerable unmet needs in providing personalized treatment options [6]. We can also find an example from the study of dementia where the informativeness of APOE ε4 as one of the best predictors of dementia varies by race [7–9]. Although useful, achieving personalization in clinical practice is challenging. Personalization requires massive data, raising privacy concerns and the potential misuse of sensitive information [10]. In this paper, we will discuss how the framework of *individualized feature selection* from machine learning (ML) can be used to efficiently guide the task of personalization in clinical assessment.

Feature selection is the process of identifying and prioritizing the most relevant and informative input variables (i.e., features) that will optimize model performance, interpretability, and generalization while minimizing model complexity and overfitting [11]. Overfitting occurs when a model becomes too complex, capturing noise in addition to the signal, which causes it to fail to generalize to unseen data (e.g., novel patients or new observations of known patients). It is important to reduce overfitting so that the model performs well in real-world scenarios [12]. This is usually achieved by using popular techniques like sequential forward selection (SFS) or backward elimination, which iteratively add or remove features to see their effect on model performance [13–16]. However, these traditional techniques lack individualization, resulting in every patient being given the same recommendation. Personalized feature selection, on the other hand, places patients at the center of the decision-making process, taking into account each individual patient's unique characteristics and recognizing that different patients may need different thresholds for diagnosis [17]. This problem definition aligns with the aim of personalized clinical assessment recommendations, where the goal is to tailor the choice of the next test based on the unique characteristics of each patient.

Recent studies in individualized feature selection have begun to address this gap by developing methods that personalize the selection of features based on individual patient data. For instance, a study on wearable electroencephalogram (EEG) monitoring platforms uses linear discriminant analysis (LDA) and the least absolute shrinkage and selection operator (LASSO) method to select discriminative features tailored to each subject's seizure patterns [18]. However, this approach does not focus on dynamic and iterative feature addition and is highly specific to EEG data. Additionally, an unsupervised personalized feature selection framework tailors feature selection to each instance in high-dimensional data [19]. However, our objective is to tackle a supervised individualized feature selection problem. Additionally, a framework employing fixed prediction models, local feature explainers, and ensembles of imputed samples provides flexible risk estimation for samples with missing features [20]. This framework relies heavily on imputations and uses a single fixed prediction model. On the other hand, we want to create a framework that provides an individualized model directly without the need for imputations or a singular prediction model.

We propose a general framework that recommends which features to obtain next for each patient, promoting a more accurate diagnosis through personalization. Taking inspiration from *locally weighted learning*, iCARE leverages patient-specific data to tailor the selection of clinical assessments for individualized healthcare recommendations used in diagnosis [21–23]. Our approach utilizes a locally weighted model tailored to each patient, which was analyzed using a feature explainer, to dynamically adapt feature selection strategies based on each patient's unique characteristics. The iCARE framework relies on three main components: (1) a sample weight calculation module, (2) an ML model trained on weighted samples, and (3) a feature explainer for the generated models. We analyzed the framework using synthetic datasets to show its personalization capability and also compared it with a traditional approach on both synthetic and real-world datasets. We hypothesized that our framework would provide more accurate diagnoses than the traditional approaches.

## 2. Methods

### 2.1. Framework architecture

Fig 1 provides an overview of the architecture of our iCARE framework. The architecture consists of an input processing module identifying missing features of incoming patients. A similarity calculation module is then used to calculate similarity scores between incoming patients and patients in the pool of known cases. This pool of known cases comprises labeled data, which includes values for predictive features such as age, sex, and test results, along with an outcome label indicating whether the individual is sick or not sick. It can be created from any data source representing known past cases with relevant features. Using these weights, a weighted logistic regression (default params: penalty = 'l2', solver = 'lbfgs', c = '1') is trained using the pool of known cases. The weights assigned to each sample reflect its relevance to the novel patient's profile, allowing for personalized model training. The trained model is then analyzed using Shapley Additive Explanations (SHAP) to quantify the importance of individual features in the locally trained logistic regression model [24]. SHAP values

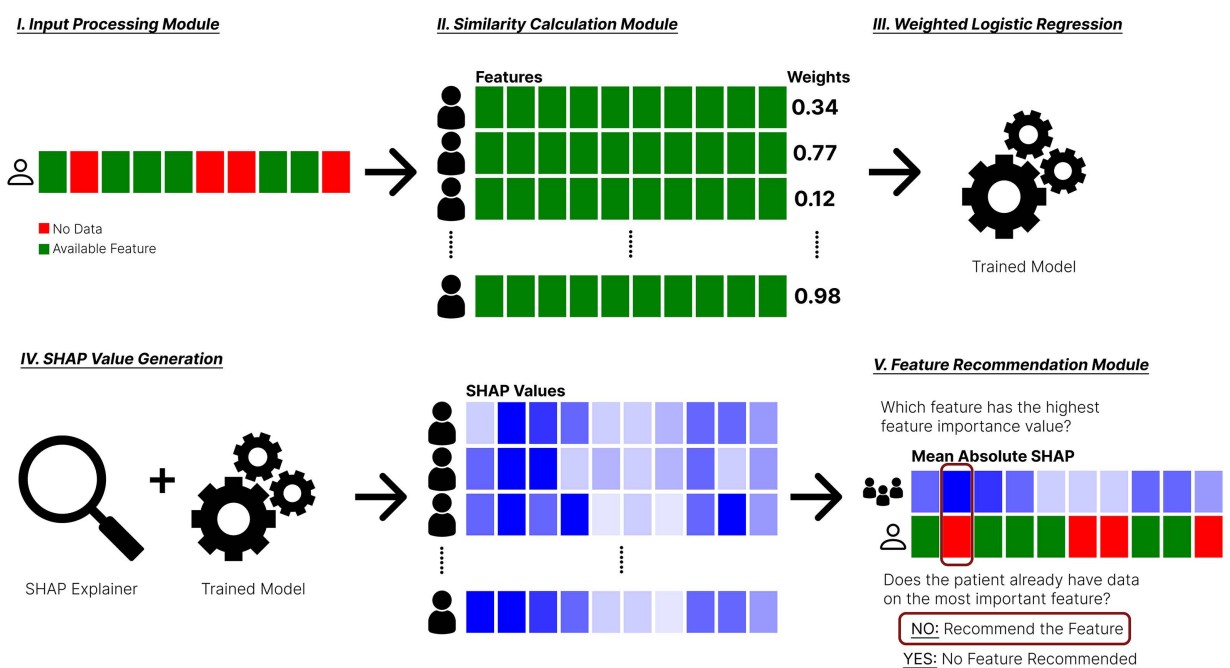

**Fig 1. Architecture of the iCARE framework.** Data were obtained from an incoming patient **(I)**, and weights were generated for the pool of known cases in the Similarity Calculation Module **(II)**. Using these sample weights, we generate a weighted logistic regression model for an incoming patient **(III)**. SHAP values are then generated using a SHAP explainer for all the subjects in the pool of known cases **(IV)**. The Feature Recommendation Module will then gather all the individual SHAP values and produce a recommendation if there is a missing feature that can be recommended to the patient **(V)**.

are based on cooperative game theory in which a prediction is broken down to show how each feature influenced the outcome of a model. SHAP treats each feature like a "player" in a game and calculates how much it helps the model make its prediction when it joins the team of features. It does this by computing the average change in the prediction when that feature is added to all possible combinations of the other features. SHAP has emerged as a tool for interpreting machine learning models in medical research, providing insights into the contribution of individual features to predictions. Studies utilizing SHAP have successfully identified key clinical features for disease prediction, such as neuropsychological test scores in Alzheimer's disease and motor and non-motor symptoms in Parkinson's disease [25,26]. In iCARE, we use a SHAP linear explainer to explain our weighted logistic regression model. For each patient, we calculate SHAP values to see how strongly each feature pushes the prediction higher or lower. By averaging these contributions, we identify which features consistently have the biggest impact. Finally, the feature recommendation module will take the explanations and produce a recommendation. It evaluates whether the feature is present in the patient's initial feature set. If any significant feature is missing, the framework recommends its inclusion to further enhance predictive accuracy.

$$sample\ weight\ =\ \frac{1}{distance} \tag{1}$$

$$Feature\ Importance_i\ =\ \frac{1}{N}\sum_{n=1}^{N}|SHAP_i| \tag{2}$$

$$Recommended\ Feature\ =\ max(Feature\ Importance) \tag{3}$$

Fig 1 illustrates the overall architecture of the iCARE framework. To complement this high-level view, Algorithms 1 and 2 provide a detailed step-by-step pseudocode representation of the framework's two core processes: calculate weight and generate iCARE feature recommendation, respectively. These algorithms formalize the logic described above and offer clarity on how the framework operates on incoming patient data.

---

**Algorithm 1:** *Calculate_Weight(dataframe, single_case, target)*

---

```
1: Get Euclidean Distance for all Samples in Dataframe:
2: distances←Euclidean_Distance(dataframe, single_case, target)
3: Convert Distances to Weights:
4: for all dist∈distances
5:   weights[i] ← 1/ (dist+1e-9)
6: return weights
```

---

This algorithm calculates the sample weights for personalized modeling. Given a single patient case and a labeled dataset, this algorithm computes the Euclidean distance between the incoming patient and all known cases. The inverse of these distances is then used to assign weights, ensuring that more similar cases exert greater influence during model training.

---

**Algorithm 2:** *Generate iCARE recommendation(dataframe, sample, target)*

---

```
1: Calculate Weights Between Each Sample in the Dataframe and the Single Sample
2: weights←Calculate_Weight(dataframe, sample, target)
3: Initialize X, y
4: X←dataframe without target column
5: y←dataframe[target]
6: Train Logistic Regression Model:
7: lr←Train Logistic Regression using X, y with sample_weight=weights
8: Find Feature Importance using SHAP:
9: explainer←SHAPLinearExplainer using lr on X
10: shap_values←generate SHAP values using explainer on X
11: for all shap_i∈shap_values
12:   shap_values[i] ← mean absolute of SHAP values in shap_i
13: Rank the Features from Most to Least Importance:
14: features←Columns of X
15: Sort features in descending order based on shap_values
16: return features
```

---

This algorithm computes patient-specific weights and uses them to train a personalized logistic regression model. SHAP is then applied to the trained model to quantify feature importance. The features are ranked based on their contribution to the model's prediction, allowing the framework to recommend the most informative features for improving predictive accuracy.

## 2.2. Experimental design

Fig 2 provides an overview of the experiment to compare the performance of the iCARE recommendation against a global feature recommendation (i.e., Global) strategy. Initially, we define a set of initial features using the *least important feature*. The dataset was split into a pool of known cases and test cases. With the procedure applied before, the test cases will have only the initial features, simulating conditions where patients don't have all the informative features. From here, we generated a global recommendation and an individualized recommendation. The global recommendation is done by training a logistic regression on the pool of known cases and analyzing it with SHAP. The feature with the highest SHAP value is selected for recommendation. On the other hand, the individualized recommendation uses the iCARE framework. We then evaluate the recommendation and repeat this process 100 times.

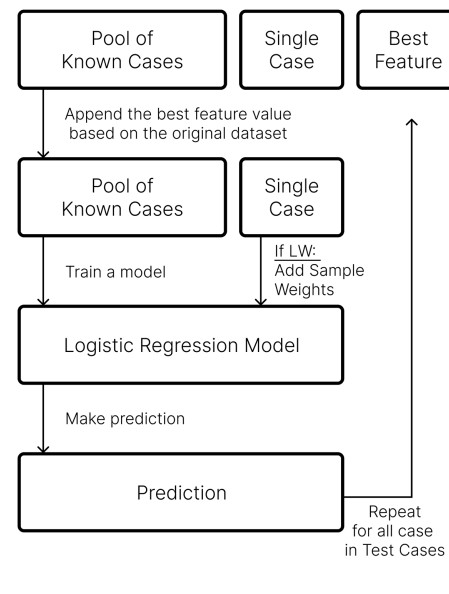

**I. Experimental Run:** *Generating Recommendation*

**II. Experimental Run:** *Evaluation*

**Fig 2. Experimental workflow to evaluate the iCARE framework.** The figure above highlights the main experimental workflow to evaluate the iCARE framework against traditional global feature selection. This workflow produces two distinct approaches to generating recommendations, as shown by the Global (i.e., global feature selection) and iCARE (i.e., individualized feature selection) split in part I. In addition, there are two distinct approaches to training the inference model, as shown in part II, where the logistic regression model can be trained with or without sample weights (i.e., LW or no LW). This produces four approaches: Global, Global+LW, iCARE, and iCARE+LW.

To evaluate the recommendations, we append the pool of known cases and a single case with the recommended feature value from the initial dataset. We then train a logistic regression using the pool of known cases and predict the outcome for the single case using the model. In addition to this, we also define the locally weighted (LW) procedure, which just uses a weighted logistic regression on this step instead of a regular logistic regression. We repeat this process until all test cases receive the predicted outcome. We then collect this prediction and calculate the accuracy and AUC (Area Under the Receiver Operating Characteristic Curve) metrics. These metrics were then averaged over 100 iterations.

This experiment was repeated using a different number of initial features. We selected the least informative feature as it represents a realistic scenario where incoming patients will more likely have less informative features. This iterative approach allowed us to assess the impact of the model performance across the various frameworks on different initial available features.

## 2.3. Dataset

We evaluate our framework with both synthetic and real-world datasets. The synthetic datasets were created to simulate ideal and non-ideal scenarios. The real-world datasets utilized in this study were obtained from the UCI Machine Learning Repository, specifically the early-stage diabetes risk prediction, heart failure clinical records, and heart disease dataset [27–29]. The early-stage diabetes dataset contains 16 features such as age, gender, polyuria, polydipsia, sudden weight loss, weakness, and other symptoms commonly associated with diabetes, along with a binary class label. The heart fail-ure dataset includes 13 clinical attributes, including age, anaemia, serum creatinine, ejection fraction, and a binary death

event label. The heart disease dataset comprises 14 features such as age, sex, chest pain type, resting blood pressure, cholesterol, fasting blood sugar, and other cardiovascular indicators, with a binary target label indicating the presence or absence of heart disease (Table 1). These datasets have been used for evaluating diagnosis using various machine learning approaches and in developing identification systems in healthcare [30–32]. We provide the code to generate the synthetic dataset, as well as the details on preprocessing steps for real-world datasets in the S1 File .

### 2.4. Statistical analysis

We performed t-tests ($\alpha = 0.05$) on the accuracy and AUC to assess the statistical significance of the performance differences between the four frameworks. To account for familywise error and reduce the risk of Type I errors, we applied Holm-adjusted p-values to the results of these multiple comparisons. Using Holm-adjusted p-values provides a more conservative and reliable measure of statistical significance compared to the standard p-values obtained from the t-test.

### 2.5. Implementation details

The iCARE framework is designed to be model-agnostic and can be used in conjunction with any machine learning algorithm suitable for the problem at hand. While this study employs logistic regression to demonstrate the framework's capacity for individualized feature selection and interpretability, it can be integrated with more complex models such as deep neural networks, random forests, or support vector machines depending on the user's needs. All experiments in this study were implemented in Python 3.11.9 using the scikit-learn library for logistic regression and the SHAP library for feature importance estimation. These experiments were conducted on a consumer-grade Windows laptop without specialized hardware (e.g., TPUs or high-performance computing).

## 3. Findings and interpretation

### 3.1. Reasoning process of the framework

The iCARE framework is grounded in the principle of localized learning and feature importance analysis to generate personalized clinical recommendations. A locally weighted logistic regression model trained using weighted patient samples from the repository of known cases focuses on learning similar patients. Due to this, iCARE will excel in scenarios where patients with similar profiles benefit from similar recommendations. Given an incoming patient with available features and a selection of potential features to be recommended, iCARE will be able to recommend the best feature given that the available features are informative of the predictiveness of the added features. For example, if in the dataset, groups of people aged below 50 benefit from additional feature A, and those above 50 benefit from additional feature B, iCARE will be able to capture this information from age (i.e., available feature) and recommend the appropriate feature (i.e., feature A or B) to an incoming patient that will give the best information gain.

We created synthetic datasets 1–3 to simulate ideal scenarios and confirm our hypothesis on the reasoning process of iCARE. Synthetic dataset 1 represents the most ideal scenario, characterized by two additional features exhibiting predictive power over different regions of the initial features value space, as shown in Fig 3. Conversely, synthetic dataset

**Table 1. Dataset size and splitting.**

| Dataset | Training Set | Testing Set | Total |
|---|---|---|---|
| Early Diabetes | 416 | 104 | 520 |
| Heart Failure | 239 | 60 | 299 |
| Heart Disease | 820 | 205 | 1025 |

The table shows the summary of dataset sizes and their train/test splits. The split follows a random 80/20 train/test split per iteration. Each dataset is listed with the number of subjects used for training, testing, and the total number of subjects.

2 illuminates the necessity for sample-weighted inference (as indicated by LW) when confronted with non-linear predictive regions highlighted in Fig 4 [33,34]. Furthermore, synthetic dataset 3 serves as a testament to the robustness of our framework, particularly in scenarios involving overlapping regions on the initial features value space that can be seen in Fig 5.

We created synthetic Datasets 4–5 to simulate hypothetical non-ideal scenarios. Synthetic Dataset 4, depicted in Fig 6, simulates a non-ideal scenario where both additional features are equally useful (i.e., the available feature does not give information about the predictiveness of the additional features). Notably, both the left and right graphs showcase identical predictive regions. This visualization emphasizes scenarios where both features share the same predictive power in the same region. In synthetic dataset 5, represented in Fig 7, we created a scenario where only one additional feature out of the rest is useful. This visualization emphasizes scenarios where one feature dominates others regarding predictive strength. The iCARE framework is expected to have similar performance to a global feature selection, highlighting no added benefit from personalization.

### 3.2. Performance on synthetic dataset

In Fig 8, we provide the comparison between the different approaches on the synthetic datasets 1–3. In synthetic dataset 1, where two additional features exhibit predictive power over distinct regions, the iCARE frameworks are expected to perform significantly better than the Global frameworks. As expected, we obtain statistically significant ($\alpha = 0.05$) differences in iCARE versus Global metrics and iCARE+LW versus Global+LW metrics using t-test, with the iCARE performing better than its Global counterpart, confirming the framework's capability to provide the best recommendation when additional features' predictive

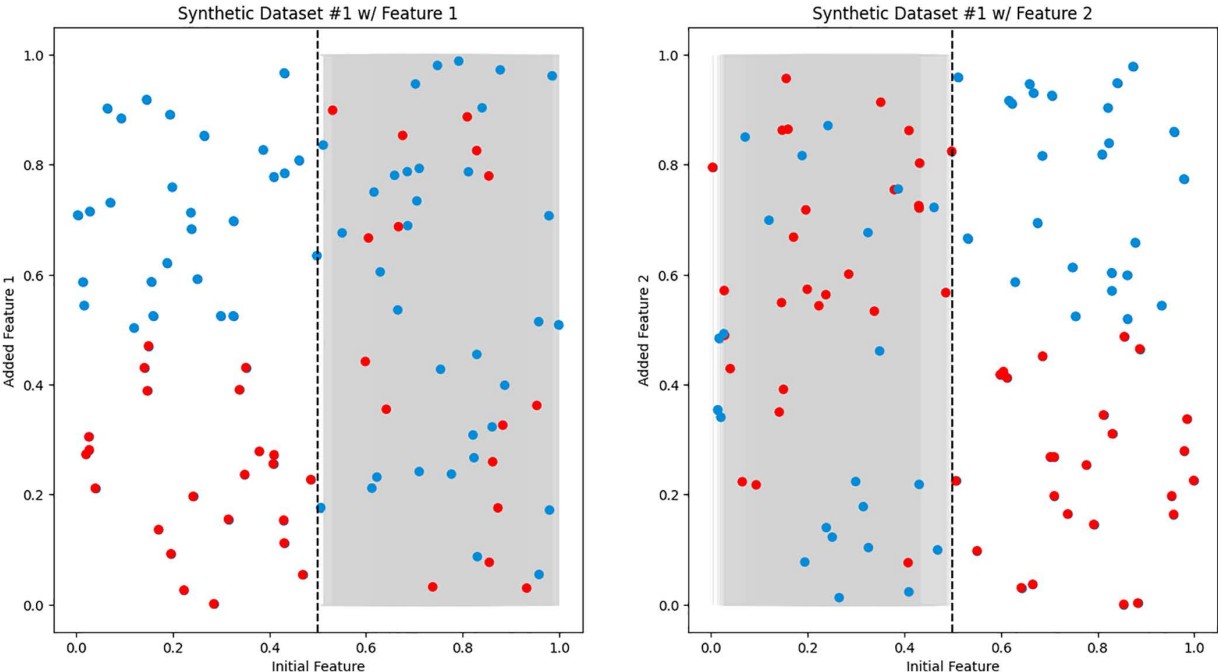

**Fig 3. Synthetic dataset 1.** Two 2D scatter plots displaying the relationship between the initial feature (x-axis) and the added feature (y-axis). The red dots represent negative samples (e.g., sick patients), while the blue dots represent positive samples (e.g., healthy patients). The left plot depicts added Feature 1, exhibiting predictive power for Initial Feature < 0.5, while random noise is observed in the shaded area above Initial Feature > 0.5. The right graph illustrates added Feature 2, demonstrating predictive power for Initial Feature > 0.5, with random noise observed in the shaded area below Initial Feature < 0.5.

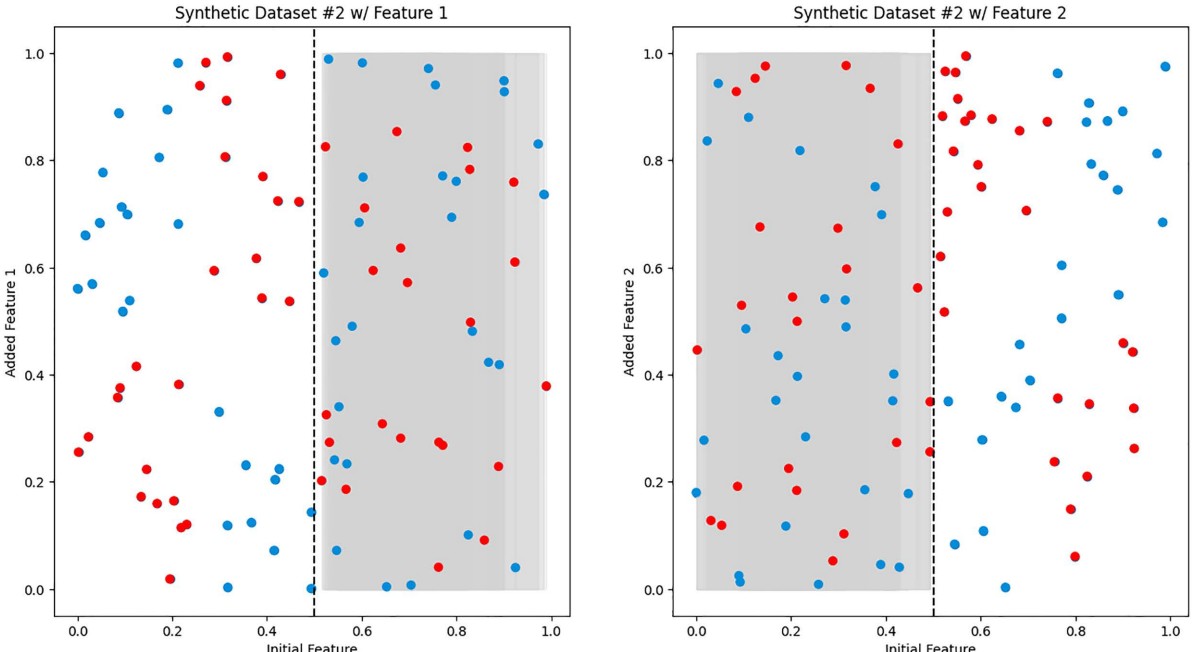

**Fig 4. Synthetic dataset 2.** Two 2D scatter plots, similar to Fig 3, showcase the relationship between the initial feature (x-axis) and the added feature (y-axis). The red dots represent negative samples (e.g., sick patients), while the blue dots represent positive samples (e.g., healthy patients). Notably, the predictive area in this dataset exhibits a non-linear pattern, suggesting a more complex relationship between the features.

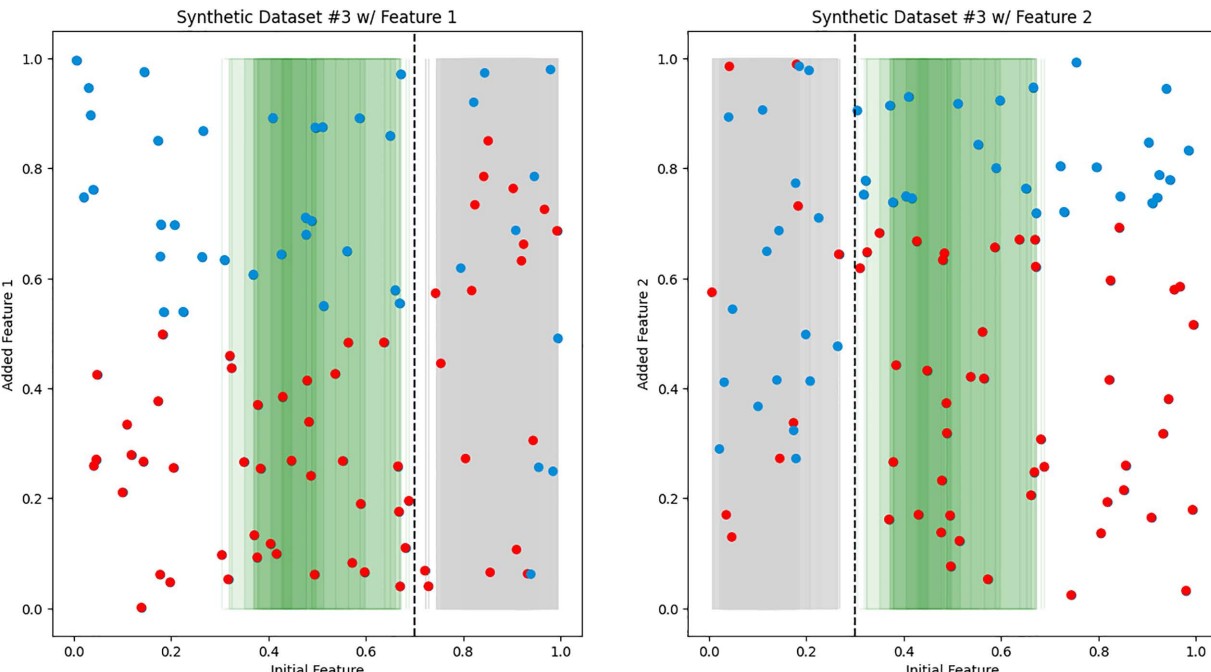

**Fig 5. Synthetic dataset 3.** 2D scatter plots resembling Fig 3, depicting the relationship between the initial feature (x-axis) and the added feature (y-axis). The red dots represent negative samples (e.g., sick patients), while the blue dots represent positive samples (e.g., healthy patients). Notably, the left graph demonstrates predictive power for $X < 0.7$, while the right graph showcases predictive power for $X > 0.3$. The green-shaded region highlights an overlapping area ($0.3 < X < 0.7$) where both features possess equal predictive power.

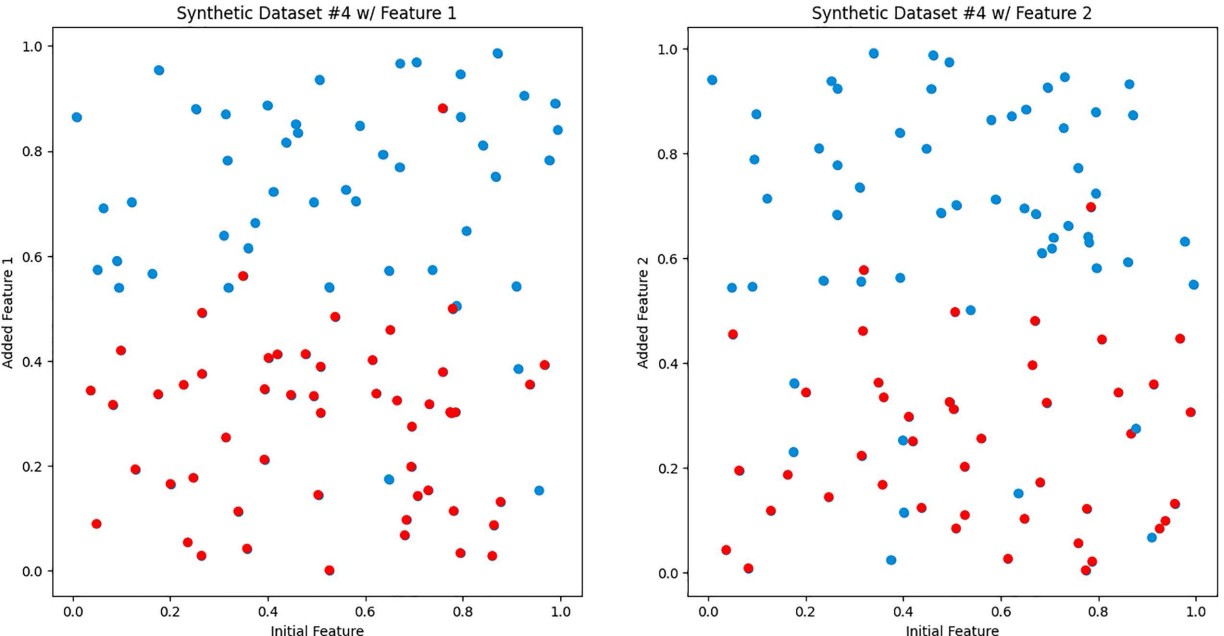

**Fig 6. Synthetic dataset 4.** Scatter plots depicting the relationship between the initial feature and the added feature, resembling the format of Fig 3. Notably, both the left and right graphs illustrate identical predictive regions.

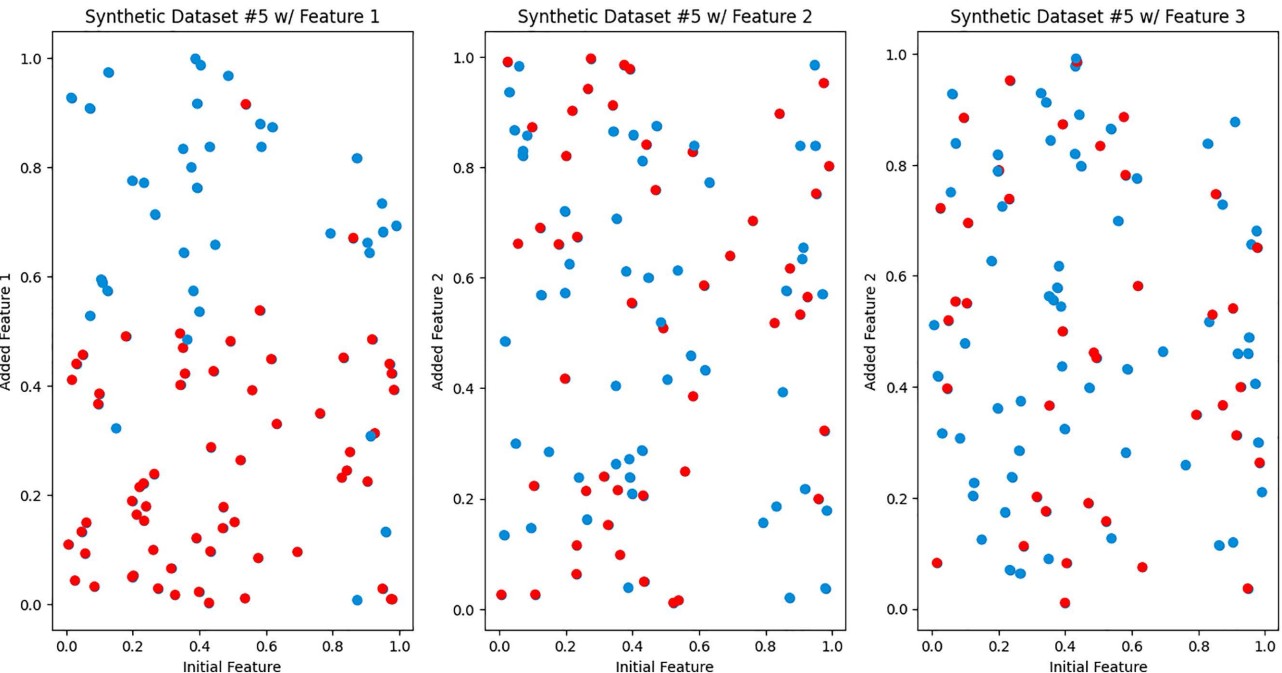

**Fig 7. Synthetic dataset 5.** Each scatter plot represents a different feature's predictive power. The first scatter plot demonstrates strong predictive capability, while the other two plots depict features with limited predictive utility. This visualization underscores the scenarios where one feature overpowers the other features.

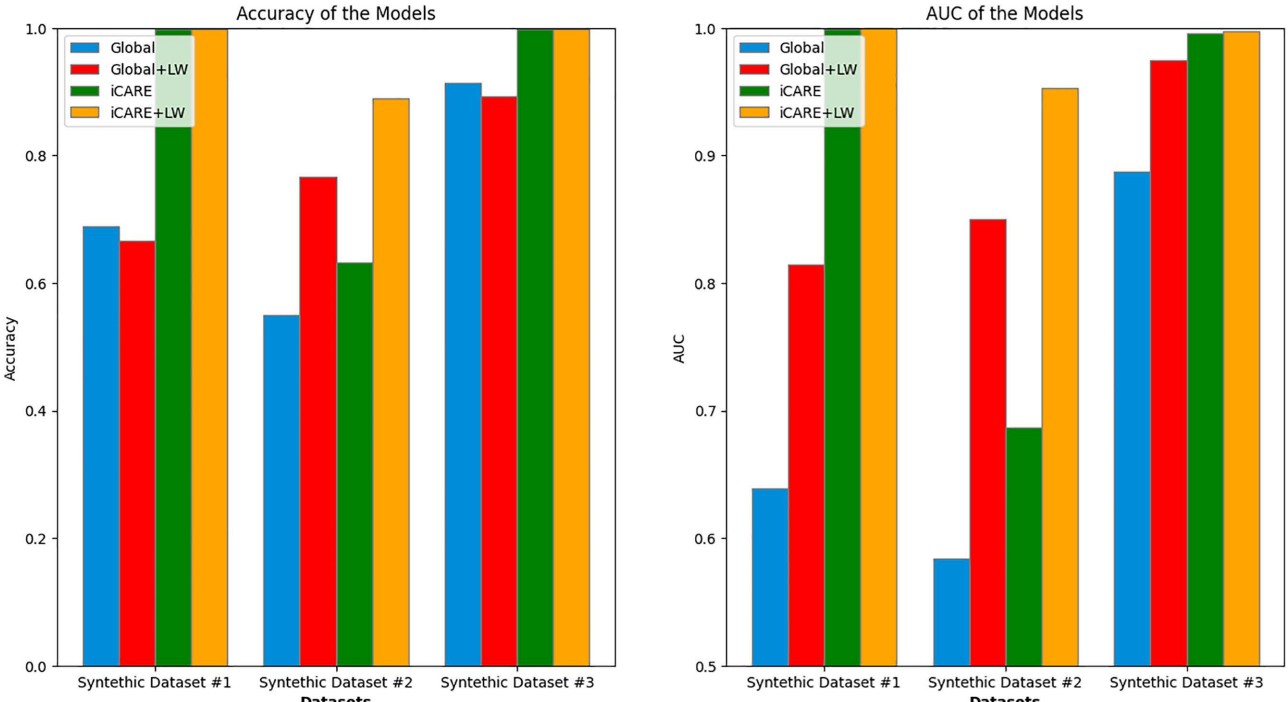

**Fig 8. Performance summary of Synthetic Dataset 1 - 3.** Comparison of accuracy (left) and area under the curve (AUC) (right) across three synthetic datasets. Each bar group represents a dataset, with values indicated for both global and local weighted metrics. For Dataset 1, the accuracy stands at 0.689, 0.667, 0.999, 0.999 with an AUC of 0.639, 0.814, 0.999, 1.0. In Dataset 2, the accuracy stands at 0.551, 0.767, 0.632, 0.891 with an AUC of 0.584, 0.850, 0.687, 0.953. Dataset 3 accuracy stands at 0.914, 0.894, 0.998, 0.998, along with an AUC of 0.888, 0.974, 0.996, 0.998. This comparison highlights variations in performance across the different synthetic datasets that represent ideal scenarios.

capabilities are clearly distinguishable given the initial feature values. Similarly, in synthetic dataset 2, characterized by non-linear predictive regions, the iCARE frameworks, especially when incorporating locally weighted inference (LW), are expected to outperform their non-LW counterparts. Statistical significance ($\alpha = 0.05$) across all comparisons can be found on our t-test, notably for iCARE versus iCARE+LW and Global versus Global+LW metrics, unseen in synthetic datasets 1 and 3. Furthermore, in synthetic dataset 3, featuring overlapping regions with identical predictive power for both features, both iCARE frameworks are expected to perform slightly better than the Global framework. The actual results align with this expectation, demonstrating the framework's ability to make accurate recommendations even in cases where features exhibit similar predictive capabilities. Similar to synthetic dataset 1, statistical significance ($\alpha = 0.05$) can be observed on our t-test when comparing iCARE with the Global framework. These results confirm the hypothesis of our framework's ability to give the best recommendation in cases where the additional features' predictive capabilities can be clearly distinguished, given the initial feature values.

In Fig 9, we provide the comparison between the different approaches on the synthetic datasets 4–5. For synthetic dataset 4, characterized by features sharing the same predictive power in the space of the initial feature value, we expected little to no difference when comparing iCARE versus Global frameworks. The actual outcome confirms this expectation, as both iCARE and Global frameworks exhibit similar performance. Similarly, for synthetic dataset 5, where there is only one useful feature, we expected a similar outcome to synthetic dataset 4. As predicted, the actual results show little variation between iCARE and Global frameworks. We observed some variances in performance; however, this can primarily be attributed to the use of locally weighted inference (i.e., LW) rather than inherent differences in the iCARE framework itself.

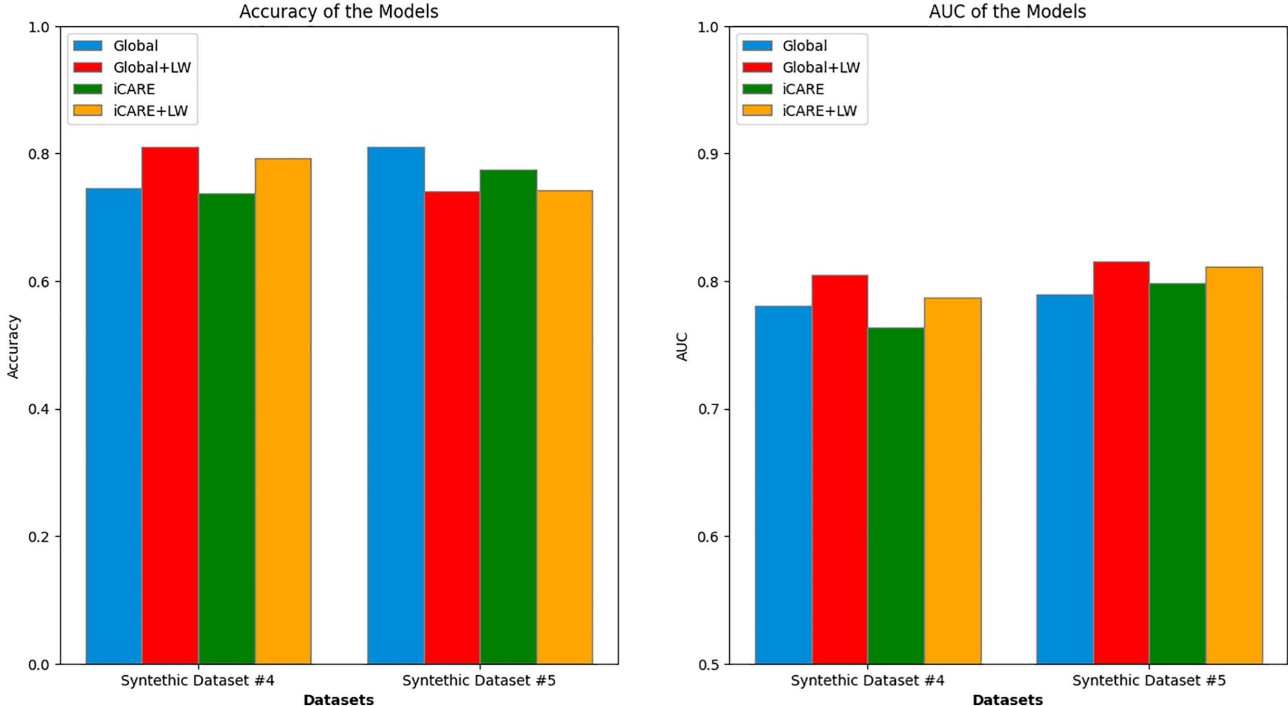

**Fig 9. Performance summary of synthetic dataset 4 - 5.** Comparison of accuracy (left) and area under the curve (AUC) (right) across Synthetic Datasets 4 and 5. Each bar group represents a dataset with performance metrics for both global and iCARE. For dataset 4, accuracy values obtained were 0.747, 0.810, 0.738, 0.792, and AUC values obtained were 0.781, 0.805, 0.764, 0.787. For dataset 5, accuracy values obtained were 0.811, 0.740, 0.774, 0.742, and AUC values obtained were 0.790, 0.815, 0.799, 0.811. These results reveal two distinct scenarios where iCARE learning fails to substantially improve global learning regarding feature addition and inference.

Furthermore, synthetic datasets 4 and 5 revealed no statistical significance for comparisons between iCARE and iCARE+LW versus Global and Global+LW metrics, which aligned with our hypothesized outcomes. The complete result of the statistical test can be seen in Table 2. These findings further confirm the hypothesis of our framework's ability to give the best recommendation in cases where the additional features' predictive capabilities can be clearly distinguished, given the initial feature values.

### 3.3. Performance on real-world dataset

In extending our evaluation to real-world scenarios, we scrutinize the performance of our framework on datasets representative of clinical contexts. Specifically, we assess its effectiveness in predicting outcomes in early diabetes and heart

**Table 2. Statistical test results of synthetic dataset 1 - 5.**

|  | Dataset 1 | | Dataset 2 | | Dataset 3 | | Dataset 4 | | Dataset 5 | |
| --- | --- | --- | --- | --- | --- | --- | --- | --- | --- | --- |
|  | ACC | AUC | ACC | AUC | ACC | AUC | ACC | AUC | ACC | AUC |
| iCARE vs Global | 0.310*** | 0.360*** | 0.082*** | 0.103*** | 0.084*** | 0.108*** | -0.009 | -0.017 | -0.036** | 0.009 |
| iCARE+LW vs Global+LW | 0.332*** | 0.186*** | 0.124*** | 0.102*** | 0.104*** | 0.023*** | -0.018 | -0.018 | 0.002 | -0.004 |
| iCARE vs iCARE+LW | 0.000 | -0.001 | -0.259*** | -0.266*** | 0.000 | -0.002 | -0.054*** | -0.023 | 0.032** | -0.013 |
| Global vs Global+LW | 0.022 | -0.176*** | -0.217*** | -0.267*** | 0.021* | -0.087*** | -0.064*** | -0.025 | 0.071*** | -0.025 |

The table shows the differences in accuracy (ACC) and area under the curve (AUC) metrics among different approaches. Specifically, it compares iCARE versus Global, iCARE+LW versus Global+LW, iCARE versus iCARE+LW, and Global versus Global+LW. Statistical significance is denoted by * for p<0.05, ** for p<0.01, and *** for p<0.001. The p-values used for testing the statistical significance above are the Holm-adjusted p-values to correct for multiple comparisons.

failure datasets, leveraging a range of personalized recommendations of features to enhance predictive accuracy and AUC metrics. In the experiment on the early diabetes dataset using three initial features, we observe that personalization leads to increased Accuracy and AUC, as seen in Fig 10. The superiority of iCARE models is shown to be statistically significant, as shown in Table 3. The three initial features that were used in this experiment are age, gender, and obesity status. Using a global approach, the feature that is recommended the majority of the time is polydipsia (i.e., excessive thirst; 75/100 iterations). It suggests that, on average, polydipsia might be more informative across the entire population when combined with age, gender, and obesity status. However, when using iCARE, two features are recommended: Polyuria (Frequent Urination) and Polydipsia. On average, Polyuria is recommended for 68% of patients, and Polydipsia is recommended for 32% of patients. The prominence of the Polyuria recommendation suggests that Polyuria might provide more relevant or discriminative information for certain patients. Polydipsia and Polyuria are both classic symptoms of diabetes [35,36]. The framework's recommendation pattern suggests variability in symptom presentation and importance among different patients. The higher recommendation of Polyuria suggests that for many patients, this symptom may be an earlier or more pronounced indicator of diabetes than Polydipsia.

In contrast, the iCARE framework does not yield substantial benefits on the heart failure dataset, as shown in Fig 11. We observed overlapping error bars in both accuracy and AUC metrics across different feature spaces in this dataset. In some instances (e.g., accuracy for the number of features = 4), iCARE models even underperform compared to their Global counterparts, highlighting the limitations of the approach in specific contexts. The statistical test in Table 4 shows that this difference in performance is statistically significant. This finding highlights a similar outcome to synthetic dataset

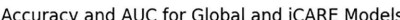

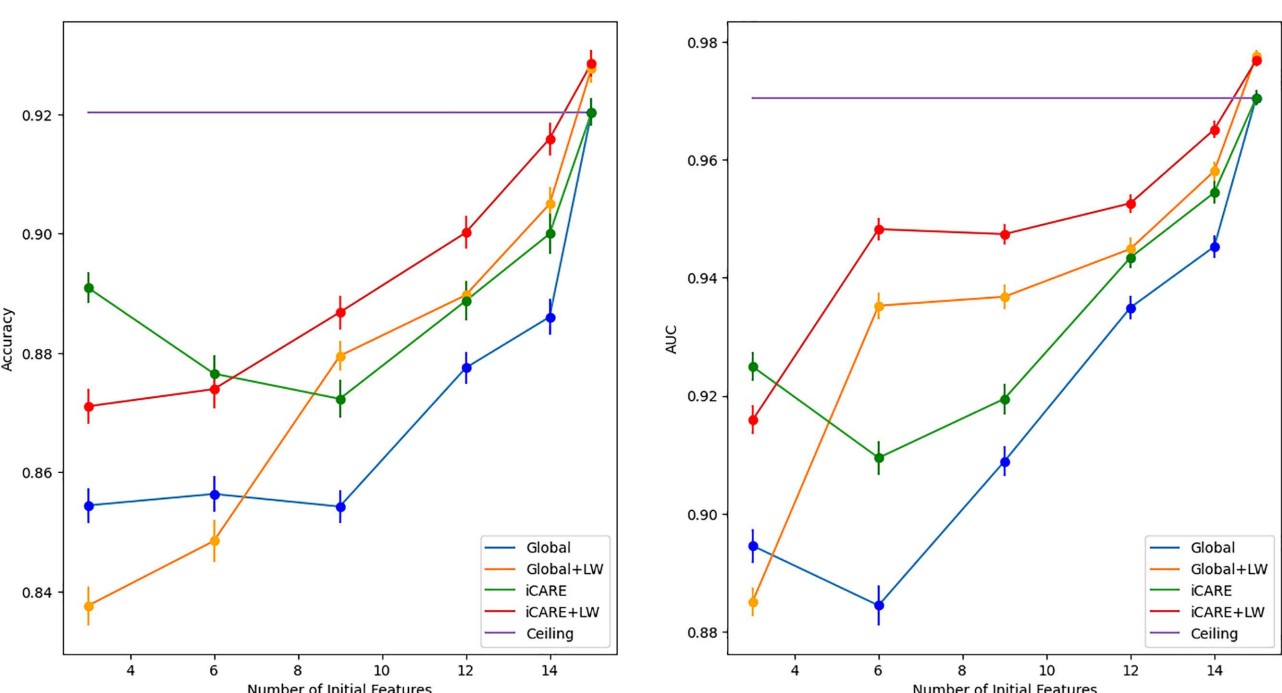

**Fig 10. Early Diabetes dataset performance summary.** This figure illustrates the mean performance of the early diabetes dataset on different feature spaces on accuracy and AUC metrics, with global and local perspectives represented by blue/orange and green/red lines, respectively. Error bars at each data point represent the standard deviation from the mean. The line graphs the maximum number of features towards the ceiling, represented by the purple line. The ceiling model represented an ML model trained on all features.

**Table 3. Early diabetes dataset performance statistical test.**

| | 3 | | 6 | | 9 | | 12 | | 14 | |
|---|---|---|---|---|---|---|---|---|---|---|
| | ACC | AUC | ACC | AUC | ACC | AUC | ACC | AUC | ACC | AUC |
| iCARE vs Global | 0.037*** | 0.030*** | 0.020*** | 0.025*** | 0.018*** | 0.011** | 0.011** | 0.008** | 0.014** | 0.009** |
| iCARE+LW vs Global+LW | 0.033*** | 0.031*** | 0.025*** | 0.013*** | 0.007 | 0.011*** | 0.010* | 0.008** | 0.011** | 0.007** |
| iCARE vs iCARE+LW | 0.020*** | 0.009* | 0.003 | -0.039*** | -0.015** | -0.028*** | -0.011* | -0.009*** | -0.016*** | -0.011*** |
| Global vs Global+LW | 0.017*** | 0.009* | 0.008 | -0.051*** | -0.025*** | -0.028*** | -0.012** | -0.010** | -0.019*** | -0.013*** |

The table shows the differences in accuracy (ACC) and area under the curve (AUC) metrics among different approaches applied to the early diabetes dataset, where the first row represents the number of initial features. Statistical significance is denoted by * for $p<0.05$, ** for $p<0.01$, and *** for $p<0.001$. The p-values used for testing the statistical significance above are the Holm-adjusted p-values to correct for multiple comparisons.

4, where it shows no added benefit when the additional features to be recommended have no distinct predictive capabilities, as well as synthetic dataset 5, where only one additional feature is useful as seen in synthetic dataset 5.

## 3.4. Comparison with other frameworks

To further evaluate the effectiveness of iCARE, we compared its performance in feature selection against the personalized imputation-based explanation-guided (Eguided) feature selection method [20]. We also provide comparisons with other global feature selection methods which are SHAP based (Global), SFS, and Lasso. As shown in Fig 12, iCARE achieved

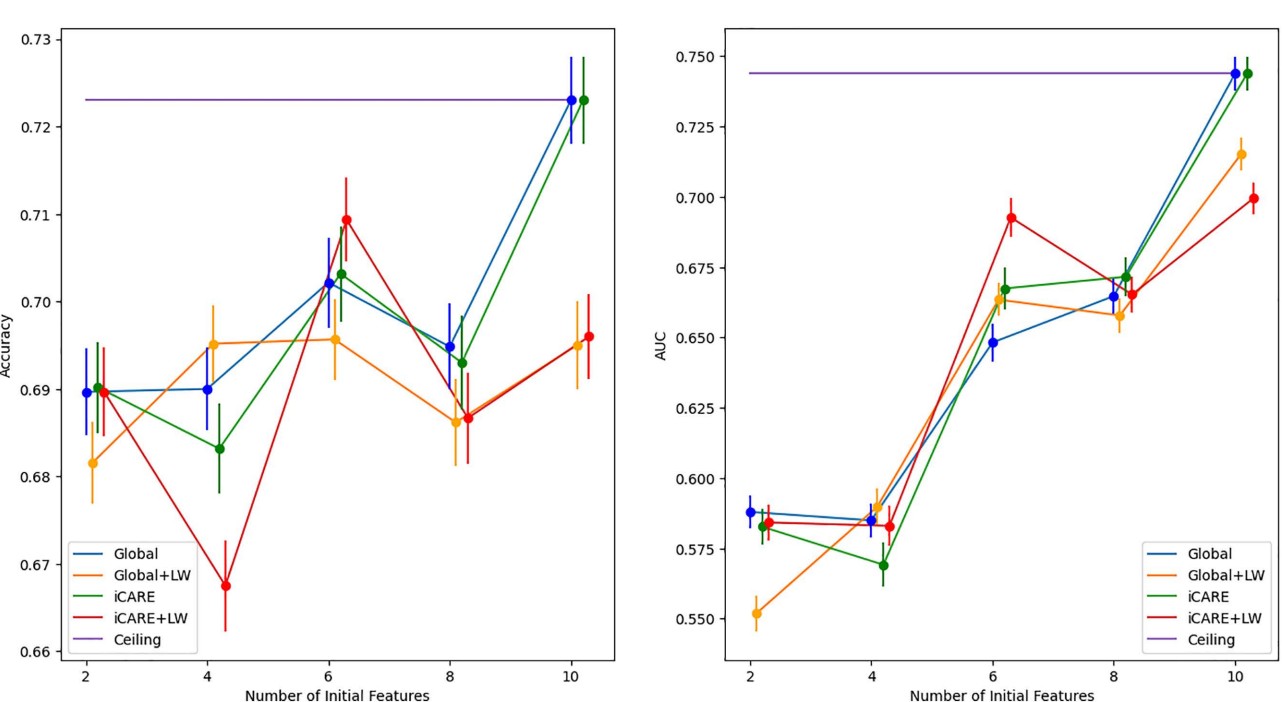

**Fig 11. Heart failure dataset performance summary.** This figure presents a comprehensive overview of mean accuracy and AUC metrics across various feature spaces on the heart failure dataset, offering insights into global and local perspectives depicted by blue/orange and green/red lines, respectively. Error bars show the standard deviation, while convergence towards the maximum features underscores notable trends.

**Table 4. Heart failure dataset performance statistical test.**

| | 2 | | 4 | | 6 | | 8 | |
|---|---|---|---|---|---|---|---|---|
| | ACC | AUC | ACC | AUC | ACC | AUC | ACC | AUC |
| iCARE vs Global | 0.001 | -0.005 | -0.007 | -0.016 | 0.001 | 0.019 | -0.002 | 0.007 |
| iCARE+LW vs Global+LW | 0.008 | 0.032*** | -0.028*** | -0.007 | 0.014 | 0.029** | 0.001 | 0.007 |
| iCARE vs iCARE+LW | 0.001 | -0.002 | 0.016 | -0.014 | -0.006 | -0.025* | 0.006 | 0.006 |
| Global vs Global+LW | 0.008 | 0.036*** | -0.005 | -0.005 | 0.006 | -0.015 | 0.009 | 0.007 |

The table shows the differences in accuracy (ACC) and area under the curve (AUC) metrics among different approaches applied to the heart failure dataset, where the first row represents the number of initial features. Statistical significance is denoted by * for $p < 0.05$, ** for $p < 0.01$, and *** for $p < 0.001$. The p-values used for testing the statistical significance above are the Holm-adjusted p-values to correct for multiple comparisons.

a +6% higher ROC-AUC score on average in the Early Diabetes Dataset and +12.1% higher ROC-AUC score on average in the Heart Disease Dataset over Eguided. Over the global feature selection methods, iCARE showed around the same increase in performance. The Heart Failure dataset highlighted before showed an example where personalized feature selection is not needed. The result showed consistency to our previous results where both iCARE and Eguided failed to perform better than a global feature selection. These results suggest that iCARE generally provides superior performance on these datasets compared to Eguided, though a few important considerations must be noted. First, the Eguided framework was originally evaluated on a much larger dataset, comprising 100,000 samples and 252 features, while our datasets are considerably smaller. Additionally, in the original Eguided study, XGBoost was employed as the prediction model, whereas we used logistic regression for both, given that iCARE relies on logistic regression. This difference in model selection may influence the relative performance of Eguided, as XGBoost could provide additional performance benefits in larger or more complex datasets. Overall, these findings reinforce the potential of iCARE as a robust feature selection framework for personalized and dynamic feature recommendations, particularly in clinical datasets of smaller scale.

### 3.5. Timing analysis

To evaluate the practicality of each feature recommendation approach in real-world scenarios, we conducted a timing analysis using two different datasets: the Early Diabetes Dataset and a synthetic dataset created for controlled experimentation. The primary objective is to assess how the number of patients requiring recommendations and the number of available features affect the computational efficiency of the methods. Importantly, our experimental setup simulates a real-world clinical setting, where patients arrive one at a time and recommendations are made on a per-patient basis. This avoids assumptions of batch processing or the possibility of precomputing feature recommendations. This setup is more realistic because in actual deployments, new patients may update the pool of known cases, invalidating any precomputed global recommendations.

The first part of the experiment investigates how the number of patients requiring feature recommendation affects the runtime. Using the Early Diabetes Dataset, we simulated a scenario where each patient starts with three known features—'Age', 'Gender', and 'Obesity'—and the remaining features are to be recommended. We varied the number of patients (N = 20, 40, 60, 80, 100), ensuring that each recommendation is performed sequentially, without batching. We plotted the result in Fig 13, the left graph, to show the timing relationship. In this setting, we observed that the Global SHAP method consistently yielded the fastest recommendation times. iCARE, Sequential Forward Selection (SFS) and LASSO required more time, and eGuided was the slowest among the methods tested.

The second part of the experiment focused on how the number of available features for recommendation influences timing. To isolate this variable, we constructed a synthetic dataset consisting of 500 samples and 100 randomly generated features, with an added binary class label. We then varied the number of features available for selection (N = 1, 20, 40, 60), again applying each recommendation method to one patient at a time. As seen in Fig 13, the right graph, Global

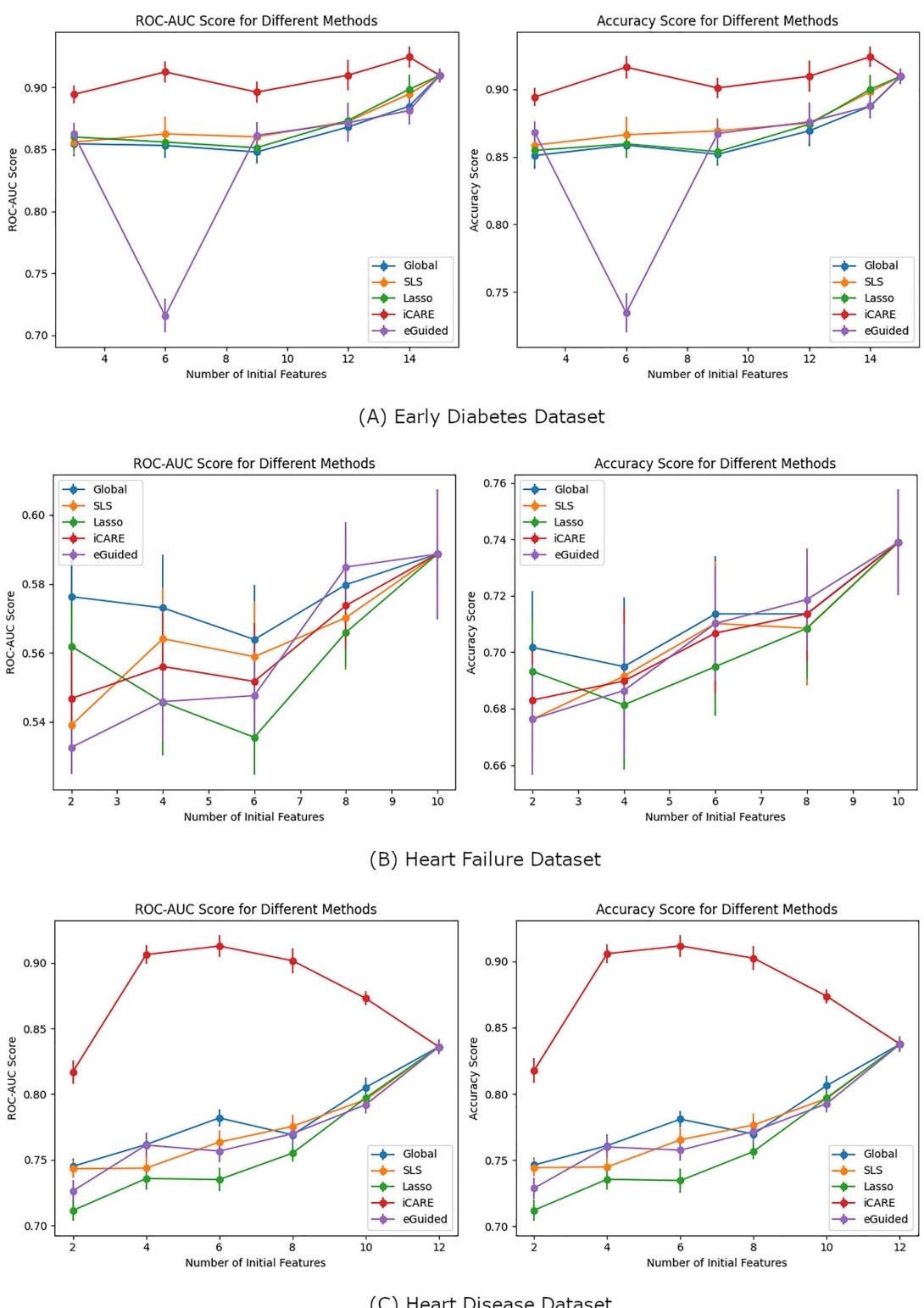

(A) Early Diabetes Dataset

(B) Heart Failure Dataset

(C) Heart Disease Dataset

**Fig 12. Performance Comparison of Feature Selection Methods Across Three Datasets.** This figure presents a comparative evaluation of AUC-ROC and Accuracy across different feature selection approaches on three real-world datasets: (A) Early Diabetes, (B) Heart Failure, and (C) Heart Disease. The graphs illustrate the performance of five feature selection methods which are Global (blue), SLS (orange), LASSO (green), iCARE (red),

and eGuided (purple), across varying feature subsets. The x-axis represents the number of selected features, while the y-axis shows the corresponding AUC-ROC and Accuracy scores, providing insights into the effectiveness of each method in optimizing predictive performance.

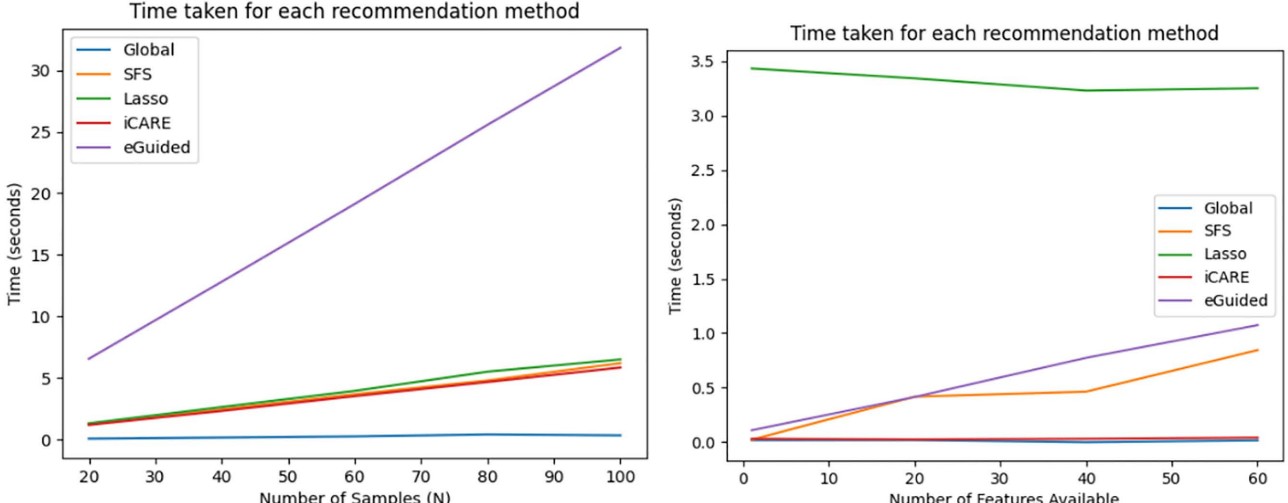

**Fig 13. Timing performance of feature recommendation methods across varying sample sizes and feature dimensions.** Runtime comparisons for five feature recommendation methods: Global SHAP, SFS, LASSO, iCARE, and eGuided, under two experimental conditions. The left panel shows how the runtime scales with the number of patients (N = 20 to 100) using the Early Diabetes Dataset, where each patient begins with three known features. The right panel displays runtime behavior as the number of available features increases (1 to 60) using a synthetic dataset of 500 samples and 100 total features.

SHAP and iCARE outperform the other methods. Their runtimes remained relatively stable despite the increasing number of features. eGuided and SFS showed moderate performance with a linear trend, LASSO demonstrated significantly higher runtimes.

These findings align well with the computational structure of each method. Global SHAP is the most efficient due to its lightweight process of training a logistic regression model and calculating SHAP values, which are fast and insensitive to both sample size and feature dimensionality. iCARE leverages SHAP for personalized recommendations, making it similarly insensitive to feature count, though it incurs additional linear time with the number of patients due to weight calculations for each sample. SFS is slower because it iteratively adds one feature at a time based on model performance, requiring multiple rounds of model training. LASSO also requires retraining and regularization over the feature space, making it computationally expensive. Lastly, eGuided is the most computationally intensive because it identifies the 100 most similar cases, imputes unknown features to generate synthetic samples, and then computes SHAP values on those samples.

## 4. Discussion

### 4.1. Importance of sample weighing

Sample weighing was utilized in the sample calculation model to create a weighted logistic regression model. This weighted model emphasizes patients with characteristics similar to those of incoming patients. The weighing strategy allows SHAP to be locally sensitive to the context of the current incoming patient, enabling feature importance rankings to be customized to the individual context rather than global trends. Sample weights have previously been used to address various challenges. For instance, a recent study proposed a weighted undersampling scheme for Support Vector

Machines (SVM) to improve classification performance in dealing with imbalanced data sets [37]. This method assigned different weights to the majority of samples based on their distance to the hyperplane, akin to how iCARE assigns weights based on patient similarity. Another research study focused on personalized diagnosis for Alzheimer's Disease, utilizing subject-specific classifiers iteratively refined through reweighting of training data [38]. Although not aimed at addressing the feature recommendation problem, the rationale for employing sample weighting remains relevant, as it serves to prioritize key subjects. Overall, incorporating sample weights in iCARE enables personalized feature rankings that can navigate diverse patient populations and complex clinical scenarios.

### 4.2. SHAP as feature importance measure

Within the iCARE framework, SHAP values play a pivotal role in selecting the most important features for personalized feature addition. We use SHAP to quantify the importance of individual features within the locally trained logistic regression model. By assigning importance values to each feature for a specific prediction, SHAP facilitates understanding the factors influencing the model's output. In the context of iCARE, SHAP integration with a weighted classifier presents a novel approach to personalized feature recommendation. This combination allows for the prioritization of features based on their impact on the current patient's prediction. While SHAP has been previously employed to measure feature importance, its integration within the framework of a weighted classifier for personalized recommendation distinguishes iCARE as a novel and impactful approach to healthcare decision support systems [39].

### 4.3. Preliminary examination of dataset

As shown in the experiment, not every dataset requires personalization. The heart failure dataset in our experiment does not benefit from our iCARE framework. We used two procedures to determine whether a dataset is suitable for personalization. The fastest dataset analysis method is to use SHAP value analysis on a ceiling model. If the analysis reveals multiple important features that contribute to the model's predictions, it suggests that the dataset may benefit from personalization. While the presence of multiple important features increases the likelihood of benefiting from personalization, it does not guarantee it. Another approach involves leveraging a pool of known cases to cross-validate the performance of the personalized model, similar to how we test our framework. While this method is slower compared to SHAP value analysis, it directly assesses the performance of personalized models through statistical testing on performance metrics accuracy and AUC values. This method confirms whether personalization is beneficial and allows us to predict how much performance gain can be expected from personalization.

### 4.4. Limitations and future directions

The iCARE framework has several limitations. First, it currently lacks a mechanism to determine whether a dataset warrants personalized feature recommendation automatically. This reliance on a naive dataset evaluation approach necessitates multiple experimental iterations, which may not be feasible in all scenarios. Future work could focus on developing robust criteria or indicators to assess the need for personalization more efficiently. Second, iCARE involves training a locally weighted model for every incoming patient, which may not be suitable for machine learning models requiring extensive training time or scenarios necessitating numerous rapid inferences. iCARE also requires a pool of known cases with a complete set of features to provide individualized feature recommendations. Moreover, iCARE assumes that the initial features available are informative of the predictive space for potential additional features, an assumption that may not always hold true. Future research should explore methods to comprehensively assess the informativeness of initial features to enhance the framework's effectiveness.

Additionally, while iCARE was evaluated on medically relevant and clinically representative datasets, these datasets do not fully substitute for real-world hospital data. To ensure the relevance of our study, we carefully selected

datasets that approximate clinical scenarios; however, further validation on real-world hospital data remains critical. We hope to explore potential collaborations to facilitate this aspect of future research and strengthen iCARE's applicability in clinical practice. Regarding privacy, future implementations will be ensured to account for compliance with regulations such as HIPAA and GDPR when incorporating real-world patient data. Furthermore, it is important to acknowledge the security risks associated with the datasets used for model training. Models trained on sensitive private data can be vulnerable to misuse or abuse, particularly in scenarios where adversaries may attempt to extract identifiable information or exploit model behavior. Future work should also consider integrating privacy-preserving techniques such as differential privacy or secure multi-party computation to protect both the data and the individuals represented in it.

Lastly, Ensuring fairness in feature selection is crucial, particularly when population representation is imbalanced. iCARE does not explicitly enforce fairness constraints, but its use of locally weighted learners inherently accounts for patient similarity, including demographic factors like race. By computing similarity scores, iCARE can identify whether an incoming patient has sufficiently comparable cases in the dataset. If not, the system can flag them as outliers, prompting caution in interpreting recommendations. Future work could build on this feature to formally incorporate fairness-aware methodologies and improve iCARE's reliability across diverse populations. Addressing these limitations and advancing research in these directions could further enhance the capabilities and applicability of the iCARE framework, ultimately contributing to improved personalized clinical assessments and decision-making in healthcare settings.

## 5. Conclusion

The iCARE system addresses the challenge of personalized feature selection in clinical assessments by dynamically tailoring the selection of clinical tests based on each patient's unique characteristics. The framework excels over a global feature selection framework in predictive accuracy, especially in cases where the initial features are informative of the predictiveness of the added features. In our experiments on early stage diabetes and heart disease datasets, iCARE demonstrated improvements of 6–12% in both accuracy and AUC compared to traditional feature selection methods. These improvements highlight the practical benefits of personalization in enhancing prediction performance, which can contribute to more accurate clinical decisions. Although personalization might not be needed in all cases, iCARE provides a flexible framework that can be applied using other machine learning algorithms. While we implemented iCARE with logistic regression in this study, the underlying framework is model-agnostic and can be adapted to work with any machine learning or deep learning model designed for classification tasks. We believe that with further testing iCARE can be applied to other domains where personalized feature recommendation can improve decision-making, such as predictive maintenance in industrial systems, financial risk assessment, or personalized learning systems in education. Future work will focus on developing automated mechanisms to determine when personalization is warranted, optimizing the framework for real-time applications, and validating its effectiveness on real-world hospital datasets to strengthen iCARE's robustness in diverse settings.

### Code and data availability

The implementation of the iCARE framework, along with all datasets used in this study, is available at the following DOI-linked repository: https://doi.org/10.5281/zenodo.15299957. This repository is a stable version of our GitHub project https://github.com/DevinRS/iCARE, and includes all source code (written in Python 3.11.9) and the dataset used in our experiments are located in the "Recreated Experiments/ExperimentData" directory.

The datasets used in this study are also publicly accessible through their original sources. The Early Stage Diabetes Risk Prediction dataset (2020) is available at https://doi.org/10.24432/C5VG8H. The Heart Failure Clinical Records dataset (2020) is available at https://doi.org/10.24432/C5Z89R. The Heart Disease Dataset compiled by David Lapp (2019) can be accessed at https://www.kaggle.com/datasets/johnsmith88/heart-disease-dataset.

### Declaration of generative AI use

During the preparation of this work the author used ChatGPT in order to improve the writing for clarity and proofreading. After using this tool the authors reviewed and edited the content as needed and take full responsibility for the content in the publication.

### Supporting information

**S1 File.** **1.** Figure 1: SHAP values bar graph for Feature 1 and Feature 2. **2.** Synthetic Dataset 1. **3.** Synthetic Dataset 2. **4.** Synthetic Dataset 3. **5.** Synthetic Dataset 4. **6.** Synthetic Dataset 5. **7.** Early Diabetes Preprocessing. **8.** Heart Failure Preprocessing.
(PDF)

### Author contributions

**Conceptualization:** Devin Setiawan, Arian Ashourvan.

**Data curation:** Devin Setiawan.

**Formal analysis:** Devin Setiawan.

**Investigation:** Devin Setiawan.

**Methodology:** Devin Setiawan, Arian Ashourvan.

**Software:** Devin Setiawan.

**Validation:** Devin Setiawan, Jeffrey M. Girrard, Arian Ashourvan.

**Visualization:** Devin Setiawan.

**Writing – original draft:** Devin Setiawan.

**Writing – review & editing:** Yumiko Wiranto, Jeffrey M. Girrard, Amber Watts, Arian Ashourvan.

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
