## [Decision Letter · Decision Letter 0]

26 Feb 2025

PDIG-D-24-00565Individualized Machine-learning-based Clinical Assessment Recommendation SystemPLOS Digital Health Dear Dr. Setiawan, Thank you for submitting your manuscript to PLOS Digital Health. After careful consideration, we feel that it has merit but does not fully meet PLOS Digital Health's publication criteria as it currently stands. Therefore, we invite you to submit a revised version of the manuscript that addresses the points raised during the review process. Please submit your revised manuscript within 60 days Apr 27 2025 11:59PM. If you will need more time than this to complete your revisions, please reply to this message or contact the journal office at digitalhealth@plos.org. Please include the following items when submitting your revised manuscript:* A rebuttal letter that responds to each point raised by the editor and reviewer(s). You should upload this letter as a separate file labeled 'Response to Reviewers '. This file does not need to include responses to any formatting updates and technical items listed in the 'Journal Requirements' section below.* A marked-up copy of your manuscript that highlights changes made to the original version. You should upload this as a separate file labeled 'Revised Manuscript with Track Changes '.* An unmarked version of your revised paper without tracked changes. You should upload this as a separate file labeled 'Manuscript '. If you would like to make changes to your financial disclosure, competing interests statement, or data availability statement, please make these updates within the submission form at the time of resubmission. Guidelines for resubmitting your figure files are available below the reviewer comments at the end of this letter. We look forward to receiving your revised manuscript. Kind regards, Sagar Barage, Ph.D.Guest EditorPLOS Digital Health Sagar BarageGuest EditorPLOS Digital Health Leo Anthony CeliEditor-in-ChiefPLOS Digital Healthorcid.org/0000-0001-6712-6626 **Journal Requirements:**

1.  We ask that a manuscript source file is provided at Revision. Please upload your manuscript file as a .doc, .docx, .rtf or .tex.

2. Please provide an Author Summary. This should appear in your manuscript between the Abstract (if applicable) and the Introduction, and should be 150–200 words long. The aim should be to make your findings accessible to a wide audience that includes both scientists and non-scientists. Sample summaries can be found on our website under Submission Guidelines:

https://journals.plos.org/digitalhealth/s/submission-guidelines#loc-parts-of-a-submission

 **Additional Editor Comments (if provided):** Dear Author,

The author used novel machine learning framework for clinical assessment of data set. However, the reviewer has raised major concern on the methodology, dataset and evaluation techniques. I recommend the article for major revision.**Reviewers' Comments:** Reviewer's Responses to Questions

**Comments to the Author**

1. Does this manuscript meet PLOS Digital Health’s publication criteria ? Is the manuscript technically sound, and do the data support the conclusions? The manuscript must describe methodologically and ethically rigorous research with conclusions that are appropriately drawn based on the data presented.

Reviewer #1: Partly

Reviewer #2: Partly

Reviewer #3: Yes

2. Has the statistical analysis been performed appropriately and rigorously?

Reviewer #1: I don't know

Reviewer #2: Yes

Reviewer #3: Yes

3. Have the authors made all data underlying the findings in their manuscript fully available (please refer to the Data Availability Statement at the start of the manuscript PDF file)?

Reviewer #1: No

Reviewer #2: Yes

Reviewer #3: Yes

4. Is the manuscript presented in an intelligible fashion and written in standard English?

Reviewer #1: Yes

Reviewer #2: Yes

Reviewer #3: Yes

5. Review Comments to the Author

Reviewer #1: Dear Authors,

Thank you for your submission. Your work on iCARE presents an innovative approach to personalized clinical assessment. I appreciate the effort in integrating locally weighted models and SHAP analysis to improve diagnostic accuracy.

However, I have a few concerns that I believe would strengthen the study:

Generalizability & Validation – Have you tested iCARE on real-world hospital datasets rather than relying solely on synthetic and publicly available data? This would enhance its clinical applicability.

Performance Benchmarking – How does iCARE compare to existing feature selection methods such as LASSO or SFS? Have you conducted ablation studies or statistical significance tests to confirm its advantages?

Reproducibility – I couldn’t find any reference to a GitHub repository or code availability. Without access to the implementation, how can the results be independently verified?

Clinical & Ethical Considerations – How does iCARE ensure fairness in feature selection across different patient demographics? Are privacy concerns (e.g., HIPAA, GDPR) addressed in the framework?

Overall, this is a promising study, but further clarification on validation, benchmarking, and reproducibility would enhance its impact. I look forward to your response and revisions.

Best regards,

Reviewer #2: 1. Describe dataset features in more details and its total size and size of (train/test) as a table and make it public.

2. Pseudocode and algorithm steps need to be inserted.

3. Time spent need to be measured in the experimental results.

4. Limitation and Discussion Sections need to be inserted.

5. All metrics need to be calculated in the experimental results.

6. The parameters used for the analysis must be provided in table

7. The architecture of the proposed model must be provided

8. Comparison with similar studies on a similar dataset need to be inserted (with references).

9. The cost associated with deploying these deep learning models, including the necessary hardware and software, is not addressed.

10. Address the accuracy/improvement percentages in the abstract and in the conclusion sections, as well as the significance of these results.

11. The authors need to make a clear proofread to avoid grammatical mistakes and typo errors.

12. Add future work in last section (conclusion) (if any)

13. Enhance the clarity of the Figures by improving their resolution.

14. The authors need to add recent articles in related work and update them.

15. To improve the Related Work and Introduction sections authors are recommended to review this highly related research work paper:

a) Secure and Transparent Lung and Colon Cancer Classification Using Blockchain and Microsoft Azure

b) Optimizing epileptic seizure recognition performance with feature scaling and dropout layers

c) Advances in ECG and PCG-based cardiovascular disease classification: a review of deep learning and machine learning methods

d) Feature reduction for hepatocellular carcinoma prediction using machine learning algorithms

e) The power of deep learning in simplifying feature selection for hepatocellular carcinoma: a review

f) Utilizing machine learning to analyze trunk movement patterns in women with postpartum low back pain

g) Employing machine learning for enhanced abdominal fat prediction in cavitation post-treatment

h) Machine learning insights into scapular stabilization for alleviating shoulder pain in college students

i) Revolutionizing core muscle analysis in female sexual dysfunction based on machine learning

j) Utilizing convolutional neural networks to classify monkeypox skin lesions

k) Predicting female pelvic tilt and lumbar angle using machine learning in case of urinary incontinence and sexual dysfunction

l) Hepatitis C Virus prediction based on machine learning framework: a real-world case study in Egypt

Reviewer #3: ### Review Report

#### **Summary of the Paper:**

The manuscript presents a novel machine-learning framework, the Individualized Clinical Assessment Recommendation System (iCARE), which employs locally weighted logistic regression and SHAP (Shapley Additive Explanations) value analysis to tailor feature selection to individual patient characteristics. The authors evaluate the framework on both synthetic and real-world datasets, demonstrating its effectiveness in enhancing diagnostic accuracy, particularly in scenarios where additional features exhibit distinct predictive capabilities. The paper is well-structured, and the methodology is sound, with clear explanations of the framework's architecture and experimental design.

---

#### **Strengths:**

1. **Innovative Approach:** The iCARE framework addresses a critical gap in clinical assessments by providing personalized feature recommendations, which is a significant advancement over traditional global feature selection methods.

2. **Comprehensive Evaluation:** The authors thoroughly evaluate the framework using both synthetic and real-world datasets, providing robust evidence of its effectiveness.

3. **Clear Methodology:** The paper provides a detailed explanation of the framework's architecture, including the use of SHAP values for feature importance, which enhances the interpretability of the model.

4. **Practical Implications:** The framework has the potential to improve diagnostic accuracy in clinical settings, particularly in personalized medicine, where individualized approaches are crucial.

---

#### **Weaknesses and Suggestions for Improvement:**

1. **Lack of Discussion on Related Work:**

- The paper would benefit from a more in-depth discussion of related work, particularly in the context of explainable machine learning (ML) and personalized medicine. Specifically, the authors should cite and discuss recent studies that have applied explainable ML to clinical data, such as:

- **D'Amore et al. (2025):** This study explores explainable ML to predict and differentiate Alzheimer's progression by sex, which is highly relevant to the iCARE framework's focus on personalized feature selection. The authors should discuss how their approach compares to or could be integrated with the methods proposed by D'Amore et al.

- **Angelini et al. (2024):** This paper uses explainable ML to unravel sex differences in Parkinson's disease, providing insights into how personalized feature selection can be applied to neurodegenerative diseases. The authors should consider how their framework could be extended to address similar challenges in Parkinson's disease or other neurological conditions.

2. **Limitations of the Framework:**

- The authors acknowledge some limitations, such as the lack of an automatic mechanism to determine whether a dataset warrants personalized feature recommendation. However, they should also discuss the potential challenges of scaling the framework to larger datasets or more complex clinical scenarios. For example, the computational cost of training locally weighted models for each patient could be prohibitive in real-time clinical settings.

- Additionally, the authors should explore the potential ethical implications of using personalized feature selection, particularly in terms of data privacy and the potential for bias in the recommendations.

3. **Comparison with Other Frameworks:**

- While the authors compare iCARE with a global feature selection approach and an imputation-based explanation-guided method, they should also consider comparing their framework with other state-of-the-art personalized feature selection methods. This would provide a more comprehensive evaluation of iCARE performance and highlight its unique contributions.

4. **Clarification on SHAP Values:**

- The authors should provide more detailed explanations of how SHAP values are calculated and interpreted within the iCARE framework. This would help readers better understand the role of SHAP in personalized feature selection and its impact on the model's recommendations.

5. **Future Directions:**

- The authors should expand on their discussion of future directions, particularly in terms of integrating iCARE with other machine learning algorithms or extending its application to other domains beyond clinical assessments. For example, they could explore how the framework could be adapted for use in predictive maintenance, financial forecasting, or other fields where personalized feature selection is valuable.

---

#### **Recommendation:**

The paper presents a novel and promising approach to personalized feature selection in clinical assessments, with strong potential for improving diagnostic accuracy. However, the manuscript would benefit from a more thorough discussion of related work, particularly in the context of explainable ML and personalized medicine, as well as a more detailed exploration of the framework's limitations and future directions.

**Recommendation:** **Accept with Major Revisions**

---

#### **Specific Suggestions for Revision:**

1. **Cite and Discuss Relevant Literature:**

- Add a discussion of **D'Amore et al. (2025)** and **Angelini et al. (2024)** in the introduction or related work section, highlighting how their approaches to explainable ML and personalized medicine relate to the iCARE framework.

- Discuss how the iCARE framework could be applied or extended to address challenges similar to those explored in these studies, such as predicting disease progression by sex or unraveling sex differences in neurodegenerative diseases.

2. **Expand on Limitations and Ethical Considerations:**

- Provide a more detailed discussion of the limitations of the iCARE framework, particularly in terms of scalability and computational cost.

- Address potential ethical concerns, such as data privacy and bias in personalized feature recommendations.

3. **Compare with Other State-of-the-Art Methods:**

- Include a comparison with other personalized feature selection methods, particularly those that have been applied in clinical or medical contexts.

- Discuss how iCARE performance compares to these methods and what unique advantages it offers.

4. **Clarify SHAP Value Calculation:**

- Provide a more detailed explanation of how SHAP values are calculated and interpreted within the iCARE framework, particularly in the context of personalized feature selection.

5. **Expand on Future Directions:**

- Discuss potential applications of the iCARE framework beyond clinical assessments, such as in predictive maintenance, financial forecasting, or other fields where personalized feature selection is valuable.

- Explore how the framework could be integrated with other machine learning algorithms or extended to address more complex clinical scenarios.

---

#### **Conclusion:**

The iCARE framework represents a significant contribution to the field of personalized medicine and clinical assessments. With the suggested revisions, particularly in terms of discussing related work and expanding on the framework's limitations and future directions, the manuscript will be strengthened and provide a more comprehensive contribution to the literature.

6. PLOS authors have the option to publish the peer review history of their article (what does this mean? ). If published, this will include your full peer review and any attached files.

**Do you want your identity to be public for this peer review?** For information about this choice, including consent withdrawal, please see our Privacy Policy .

Reviewer #1: **Yes: ** akbar ali

Reviewer #2: No

Reviewer #3: No

---

## [Decision Letter · Decision Letter 1]

3 Sep 2025

Individualized Machine-learning-based Clinical Assessment Recommendation System

PDIG-D-24-00565R1

Dear Dr. Setiawan,

We're pleased to inform you that your manuscript has been judged scientifically suitable for publication and will be formally accepted for publication once it meets all outstanding technical requirements.

Within one week, you'll receive an e-mail detailing the required amendments. When these have been addressed, you'll receive a formal acceptance letter and your manuscript will be scheduled for publication.

An invoice for payment will follow shortly after the formal acceptance. To ensure an efficient process, please log into Editorial Manager at https://www.editorialmanager.com/pdig/ click the 'Update My Information' link at the top of the page, and double check that your user information is up-to-date. For billing related questions, please contact billing support at https://plos.my.site.com/s/.

Kind regards,

Martin G Frasch

Section Editor

PLOS Digital Health

Additional Editor Comments (optional):

Dear Author,

we have received comment on your research article "Individualized Machine-learning-based Clinical Assessment Recommendation System". We request you to please address all comments raised by potential reviewer.

Reviewers' comments:

Reviewer's Responses to Questions

**Comments to the Author**

1. If the authors have adequately addressed your comments raised in a previous round of review and you feel that this manuscript is now acceptable for publication, you may indicate that here to bypass the “Comments to the Author” section, enter your conflict of interest statement in the “Confidential to Editor” section, and submit your "Accept" recommendation.

Reviewer #2: All comments have been addressed

Reviewer #4: All comments have been addressed

2. Does this manuscript meet PLOS Digital Health’s publication criteria ? Is the manuscript technically sound, and do the data support the conclusions? The manuscript must describe methodologically and ethically rigorous research with conclusions that are appropriately drawn based on the data presented.

Reviewer #2: (No Response)

Reviewer #4: Yes

3. Has the statistical analysis been performed appropriately and rigorously?

Reviewer #2: (No Response)

Reviewer #4: Yes

4. Have the authors made all data underlying the findings in their manuscript fully available (please refer to the Data Availability Statement at the start of the manuscript PDF file)?

Reviewer #2: (No Response)

Reviewer #4: Yes

5. Is the manuscript presented in an intelligible fashion and written in standard English?

PLOS Digital Health does not copyedit accepted manuscripts, so the language in submitted articles must be clear, correct, and unambiguous. Any typographical or grammatical errors should be corrected at revision, so please note any specific errors here.

Reviewer #2: (No Response)

Reviewer #4: Yes

6. Review Comments to the Author

Please use the space provided to explain your answers to the questions above. You may also include additional comments for the author, including concerns about dual publication, research ethics, or publication ethics. (Please upload your review as an attachment if it exceeds 20,000 characters)

Reviewer #2: Accept.

Reviewer #4: The authors have addressed most of the concerns from previous reviewers. I don't have any additional comments.

7. PLOS authors have the option to publish the peer review history of their article (what does this mean? ). If published, this will include your full peer review and any attached files.

**Do you want your identity to be public for this peer review?** For information about this choice, including consent withdrawal, please see our Privacy Policy . 

Reviewer #2: None

Reviewer #4: No
